# Vascular cell-adhesion molecule 1 (VCAM-1) regulates JunB-mediated IL-8/CXCL1 expression and pathological neovascularization

Geetika Kaur [1,2], Deepti Sharma[1,2], Shivantika Bisen[1,2], Chandra Sekhar Mukhopadhyay[3], Katherine Gurdziel[4] & Nikhlesh K. Singh [1,2 ✉]

Vascular adhesion molecules play an important role in various immunological disorders, particularly in cancers. However, little is known regarding the role of these adhesion molecules in proliferative retinopathies. We observed that IL-33 regulates VCAM-1 expression in human retinal endothelial cells and that genetic deletion of IL-33 reduces hypoxia-induced VCAM-1 expression and retinal neovascularization in C57BL/6 mice. We found that VCAM-1 via JunB regulates IL-8 promoter activity and expression in human retinal endothelial cells. In addition, our study outlines the regulatory role of VCAM-1-JunB-IL-8 signaling on retinal endothelial cell sprouting and angiogenesis. Our RNA sequencing results show an induced expression of CXCL1 (a murine functional homolog of IL-8) in the hypoxic retina, and intravitreal injection of VCAM-1 siRNA not only decreases hypoxia-induced VCAM-1-JunB-CXCL1 signaling but also reduces OIR-induced sprouting and retinal neovascularization. These findings suggest that VCAM-1-JunB-IL-8 signaling plays a crucial role in retinal neovascularization, and its antagonism might provide an advanced treatment option for proliferative retinopathies.

[1] Integrative Biosciences Center, Wayne State University, Detroit, MI 48202, USA. [2] Department of Ophthalmology, Visual and Anatomical Sciences, School of Medicine, Wayne State University, Detroit, MI 48202, USA. [3] School of Animal Biotechnology, Guru Angad Dev Veterinary and Animal Sciences University, Ludhiana, Punjab 141004, India. [4] Institute of Environmental Health Sciences and Department of Pharmacology, School of Medicine, Wayne State University, Detroit, MI 48202, USA. ✉email: nsingh2@wayne.edu

Retinal neovascularization is an underlying pathological cause of vision impairment and blindness in retinal vein occlusion (RVO), retinopathy of prematurity (ROP), wet age-related macular degeneration, and diabetic retinopathy (DR). These differential ocular diseases are linked with different age groups[1]. ROP with retinal neovascularization primarily afflicts preterm infants and causes blindness in children. During retinal neovascularization, abnormal blood vessels extend to the retinal surface and vitreous by forming sprouts and branches, resulting in neovascularization[2]. Conventional therapeutic regimens for these diseases include laser ablation and anti-VEGF therapies[1,3]. The anti-VEGF therapies are effective but are often associated with macular edema, reduced night vision, tractional detachment, and atrophy[4–6]. Therefore, considering the clinical issues related to the anti-VEGF therapies, investigation of other effective mediator(s) could potentially open advanced therapeutic avenues in neovascularization-associated retinopathies.

Several angiogenic cytokines, interleukins, growth factors, adhesion molecules, and inflammatory mediators have been implicated in the progression of ocular diseases[7,8]. Adhesion molecules such as intracellular cell adhesion molecules (ICAM-1), and vascular cell adhesion molecule (VCAM-1) are suggested as the main drivers of neovascularization in diabetic retinopathy (DR)[9]. VCAM-1, an inducible transmembrane glycoprotein is mostly expressed in endothelial cells[10,11]. Various pro-inflammatory cytokines, reactive oxygen species (ROS), high glucose concentration, shear stress, toll-like receptor agonists, and oxidized low-density lipoprotein induce VCAM-1 expression[12]. VCAM-1 consists of a cytosolic domain, a transmembrane domain, and an extracellular domain[13]. A study has shown that VCAM-1 plays a pro-angiogenic role in oxidative stress-induced neovascularization. During inflammatory responses, binding of ligands to VCAM-1 activates endothelial cells leading to activation of calcium fluxes and Rac1, which further activates nicotinamide adenine dinucleotide phosphate (NADPH) oxidase 2, resulting in ROS generation[14]. It was also shown that hyperglycemia and hyperlipidemia induce the expression of VCAM-1 in mice retinal vessels[15]. Many studies have suggested the relevance of VCAM-1 in angiogenesis. It has been reported that serum VCAM-1 level correlates with the microvessel density, and it was suggested that serum VCAM-1 level might be used as a marker for angiogenesis in breast cancer[16]. This report was further backed by the findings that overexpression of VCAM-1 induces microvessel density in gastric cancer[17]. It was also suggested that VCAM-1 and α4β1 integrin (VLA-4) induces angiogenesis by promoting the intracellular adhesion between endothelial cells and pericytes. Furthermore, the administration of anti-VCAM-1 antibody reduces matrigel plug angiogenesis in murine models[18]. It was also shown that exposure to anti-VCAM-1 antibody attenuated IL-4 and IL-13-induced tube formation in human microvascular endothelial cells[19]. Although studies have suggested that VCAM-1 might have a role in angiogenesis, nothing is known regarding the direct role of VCAM-1 on angiogenic signaling and pathological retinal neovascularization.

IL-33 belongs to IL-1 family of cytokines and is an emerging regulator of inflammatory diseases. It is mainly expressed in epithelial cells, endothelial cells, and fibroblasts. Studies from our lab and others have shown that IL-33 induces angiogenesis in endothelial cells[20,21]. Contrary to it, few reports have also reported that IL-33 attenuates ocular angiogenesis[22]. Contrasting reports are present in the literature suggesting a proangiogenic and anti-angiogenic role of IL-33. However, the role of IL-33 in angiogenesis remains obscure. Based on this background, the present study aimed to evaluate the functional significance of IL-33 induced VCAM-1 signaling on pathological angiogenesis in hypoxic/ischemic retinopathies.

## Results

**Expression of VCAM-1 is increased in ischemic retina.** Retinal neovascularization is a clinical manifestation of retinopathy of prematurity, diabetic retinopathy, and other ischemic/hypoxic retinopathies. Retinal ischemia is a common attribute that leads to retinal neovascularization in these retinopathies. It has also been shown that oxidative stress-induced expression of endothelial cell adhesion molecules promotes a proangiogenic environment in the eye leading to retinal neovascularization[23]. Therefore, we looked for the expression of endothelial cell adhesion molecules in the murine model of oxygen-induced retinopathy (OIR) (Fig. 1a). In the OIR model, mouse pups are exposed to hyperoxia (75% oxygen) from P7 to P12, resulting in regression and cessation of normal radial vessels. This cascade of events mimics the first phase of retinopathy of prematurity (ROP). From P12 to P17, these pups are returned to ambient air (normoxia), which leads to the generation of relative hypoxia in nonperfused regions of the retina, which results in retinal neovascularization. This phase of OIR emulates certain outcomes of proliferative diabetic retinopathy, and is comparable to the second phase of ROP.

The retinal tissue of C57BL/6 mice exposed to OIR was collected from various periods of relative hypoxia and analysed for various endothelial adhesion molecules (VCAM-1, ICAM-1 and E-selectin). We observed an induced expression of VCAM-1 in the hypoxic retina at all time points as compared to normoxia (Fig. 1b). However, no significant regulation was observed in the expression of ICAM-1 in the hypoxic retina. We have previously shown that OIR induces IL-33 expression in hypoxic mice retina, and a study has reported a role for IL-33 in atherosclerotic lesion development via promoting MCP-1 and adhesion molecules expression in the endothelium[24]. Therefore, we next studied the effect of IL-33 on endothelial adhesion molecules expression in human retinal microvascular endothelial cells (HRMVECs). The VCAM-1 expression was induced by IL-33 in HRMVECs (Fig. 1c). We next generated IL-33 knockout mice to understand the role of IL-33 deletion on VCAM-1 expression in retina (Fig. 1d). The genetic deletion of IL-33 leads to a complete loss of IL-33 levels in the whole retinal lysate (Fig. 1e). Furthermore, we also observed that depletion of IL-33 reduces OIR-induced VCAM-1 levels in hypoxic retina (Fig. 1f). Increased vascular endothelial growth factor (VEGF) expression has been observed in the retinas of mice subjected to OIR[25], and VEGF has been shown to regulate VCAM-1 expression in the retina[26]. As a result, we looked for the role of IL-33 in OIR-induced VEGF levels in mice retinas and observed that IL-33 depletion has no significant effect on OIR-induced VEGF levels (Fig. 1g). Our previous study showed that genetic deletion of IL-33 reduces retinal neovascularization in C57BL/6 mice[21]. A study reported significantly higher vitreous levels of vascular cell adhesion molecule (VCAM-1) in patients with proliferative diabetic retinopathy[27]. Therefore, understanding the direct involvement of VCAM-1 controlled signaling in pathological retinal neovascularization may thus offer up advanced therapeutic approaches for proliferative retinopathies.

**NF-κB mediated VCAM-1 expression regulates IL-33-induced angiogenic events in human retinal endothelial cells.** Recent reports have shown that VCAM-1 levels are associated with tumor angiogenesis and metastasis[28]. To decipher the functional significance of VCAM-1 levels in the hypoxic retina, we next investigated the role of VCAM-1 on IL-33 induced sprouting, migration, and tube formation of HRMVECs. The effect of VCAM-1 on IL-33-induced tip cell formation and tube formation was evaluated using a 3 dimensional (3-D) sprouting assay

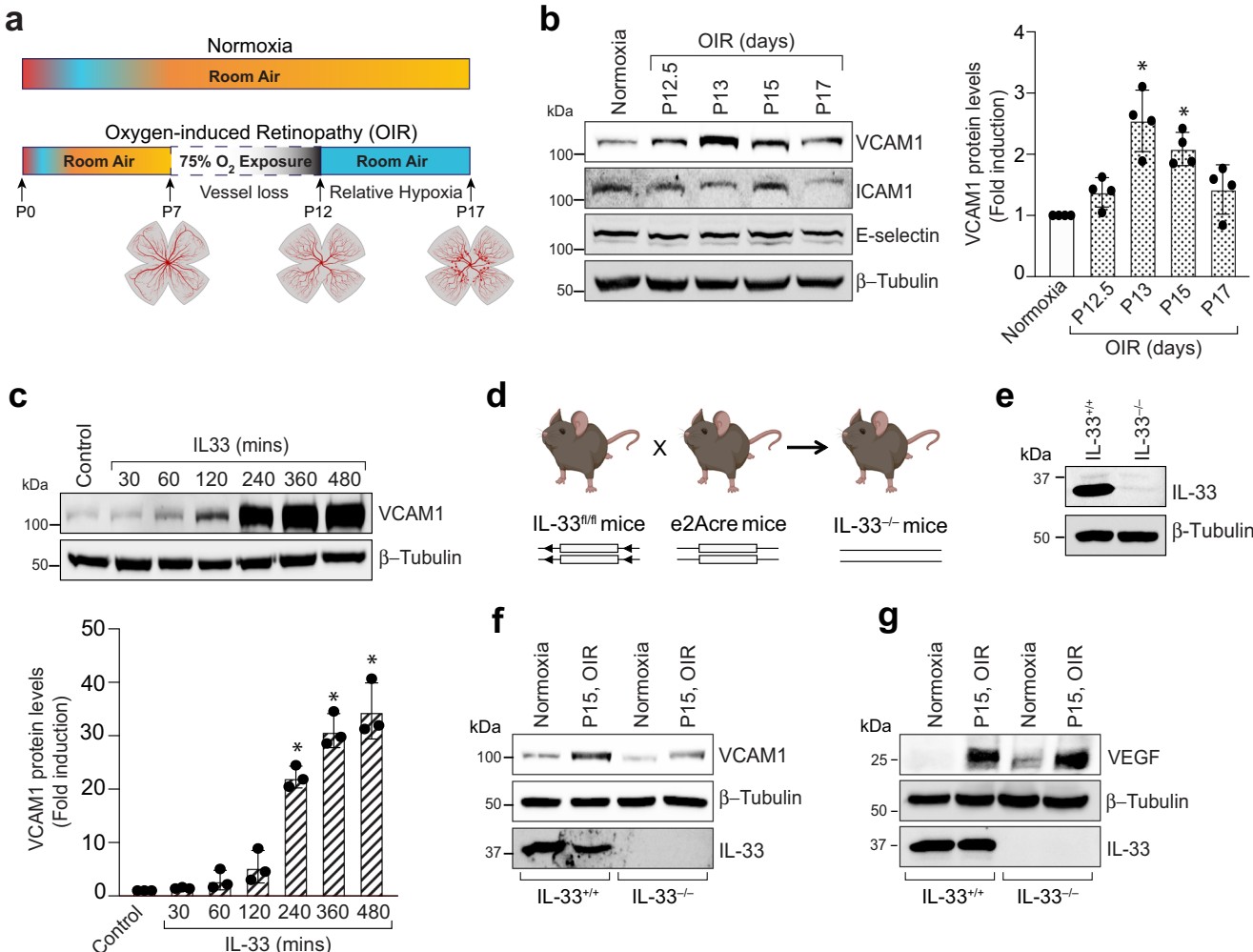

**Fig. 1 OIR induces VCAM-1 levels in retinal endothelial cells. a** Schematic illustration describing the murine OIR model. The C57BL/6 mice pups were exposed to 75% oxygen from P7 to P12, and then returned to room air. **b** At various periods of relative hypoxia, eyes were enucleated, retinas isolated, tissue extracts made and analyzed by Western blotting for VCAM-1, ICAM1, and E-selectin levels using their specific antibodies and normalized to β-tubulin. The bar graph in panel (**b**) show the quantitative analysis of 4 independent experiments ($n = 4$), expressed as Mean ± SD. **c** an equal amount of protein from control and the indicated time periods of IL-33 (20 ng/ml) treated HRMVECs were analyzed by Western blotting for VCAM-1 levels and normalized to β-tubulin. The bar graph in panel (**c**) show the quantitative analysis of 3 independent experiments ($n = 3$), expressed as Mean ± SD. **d** A Schematic diagram of breeding strategy for the generation of IL-33 knockout mice. **e** Eyes from IL-33$^{+/+}$ and IL-33$^{-/-}$ pups were enucleated, retinas isolated, and extracts were analyzed for IL-33 levels by Western blotting and normalized to β-tubulin. **f, g** At P15 (post 72-hour hypoxia), retinal extracts from IL-33$^{+/+}$ and IL-33$^{-/-}$ mice pups were analyzed by Western blotting for indicated proteins and normalized to β-tubulin. *$P < 0.05$ vs normoxia or control. Figure 1a & d were created with BioRender.com.

and growth factor reduced matrigel assay, respectively. The modified Boyden Chamber and endothelial cell wound healing assay was used to measure HRMVECs migration. VCAM-1 depletion using its siRNA attenuated IL-33 induced migration, wound healing, tube formation, and sprouting of HRMVECs (Fig. 2a–e).

The role of NF-κB signaling has been implicated in LPS-induced VCAM-1 expression in Human aortic endothelial cells[29]. Therefore, we looked for the role of NF-κB signaling on IL-33-induced VCAM-1 expression in HRMVECs. IL-33 induces the phosphorylation of IKKα/β, IκBα, and NF-κB in HRMVECs (Fig. 2f), and blockade of NF-κB activation by QNZ (4-N-[2-(4-Phenoxyphenyl) ethyl]-1,2-dihydroquinazoline-4,6-diamine) and PDTC (Pyrrolidine dithiocarbamate ammonium), attenuated IL-33-induced VCAM-1 expression in HRMVECs (Fig. 2g & h). We also observed that hypoxia induced NF-κB activation in retina, and genetic depletion of IL-33 reduced NF-κB activation in retina (Fig. 2i).

**VCAM-1 regulates IL-33-induced expression of angiogenic protein IL-8.** To understand the VCAM-1 signaling that regulates angiogenesis in HRMVECs, we used a protein profiler human angiogenesis array kit. For this, HRMVECs were transfected with control and VCAM-1 siRNA and treated with IL-33 for 8 h. We observed an increased expression of IL-8 by IL-33 and downregulation of VCAM-1 level with its siRNA attenuated IL-8 expression in HRMVECs (Fig. 3a). To confirm our observation, we evaluated the effect of IL-33 on IL-8 expression, both at mRNA and protein levels in HRMVECs. We observed a significant increase in IL-8 expression at both mRNA and protein levels by IL-33 in HRMVECs (Fig. 3b & c). In addition, siRNA-mediated depletion of VCAM-1 levels attenuated IL-33-induced IL-8 expression at both mRNA and protein levels in HRMVECs (Fig. 3d & e). We also observed that VCAM-1 depletion has no effect on VEGF levels in HRMVECs (Fig. 3e). Furthermore, NF-κB inhibitors QNZ and PDTC also blocked IL-33-induced

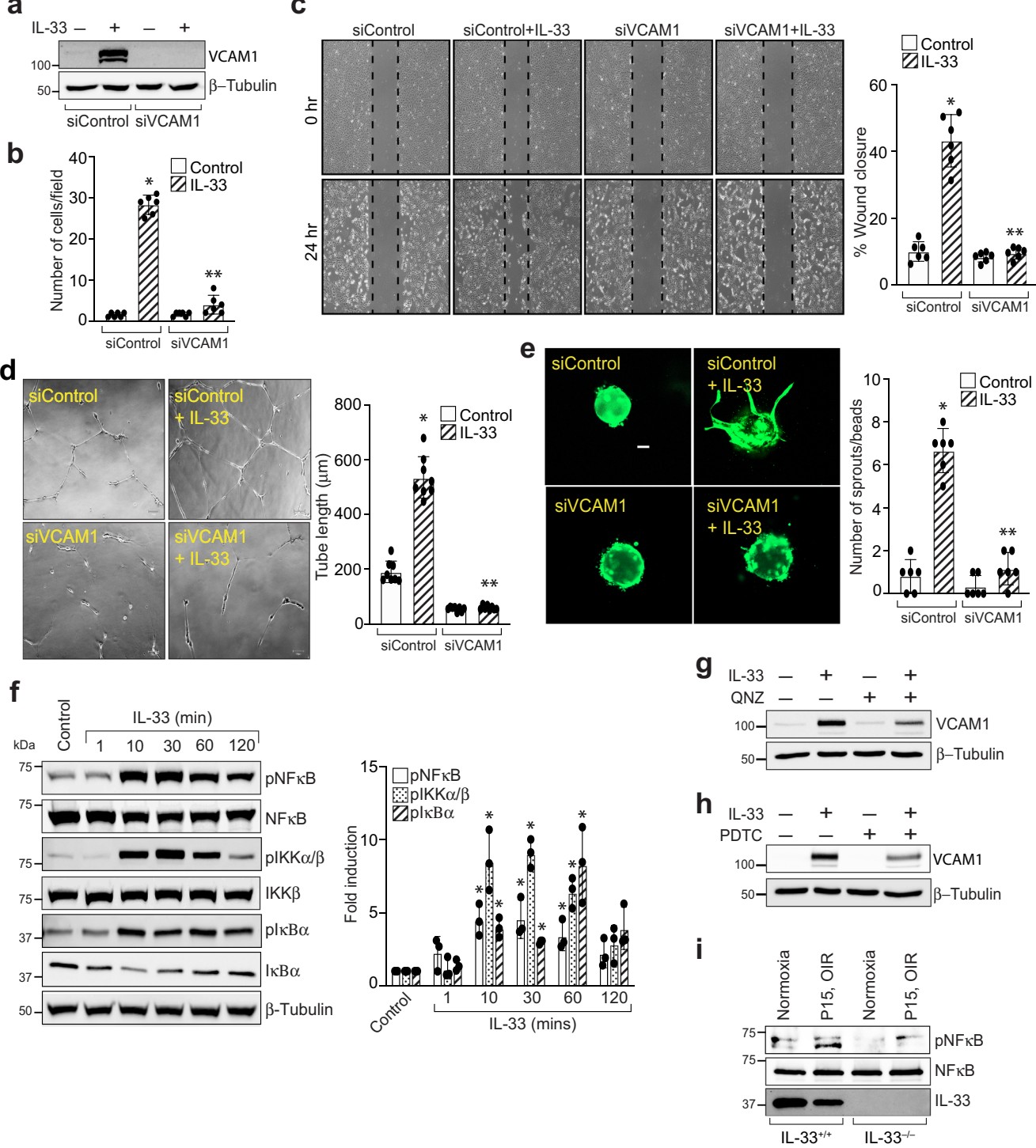

**Fig. 2 VCAM-1 mediates IL-33-induced angiogenic events in HRMVECs. a** HRMVECs were transfected with control siRNA (siControl) or VCAM-1 siRNA (siVCAM-1), after 48 h cell extracts were prepared and analyzed for VCAM-1 levels by Western blotting and normalized to β-tubulin. **b–e** All the condition are same as in panel a, except that the cells were analyzed for IL-33 (20 ng/mL)-induced migration (**b**), wound healing (**c**), tube formation (**d**), and sprouting assay (**e**). **f** Cell extracts treated with various time periods of IL-33 (20 ng/mL), and were analyzed by Western blotting for the indicated proteins. **g, h** Quiescent HRMVECs were first treated with QNZ (10 μM, NFκB inhibitor) or PDTC (10 μM, NFκB inhibitor) for 30 min, followed with or without IL-33 (20 ng/mL) treatment for 8 h. The cell lysates were prepared and analysed for VCAM-1 levels by Western blotting and normalized to β-tubulin. **i** At P15, the retinal tissue extracts from IL-33[+/+] and IL-33[−/−] mice pups subjected to normoxia and hypoxia were analyzed for indicated proteins by Western blotting. $n = 6$ (**b, c, d, e**) or $n = 3$ (**f**) biologically independent samples per group, expressed as Mean ± SD. *$P < 0.05$ vs control or siControl, **$p < 0.05$ vs siControl + IL-33. Scale bar represents 50 μm in panel (**e**).

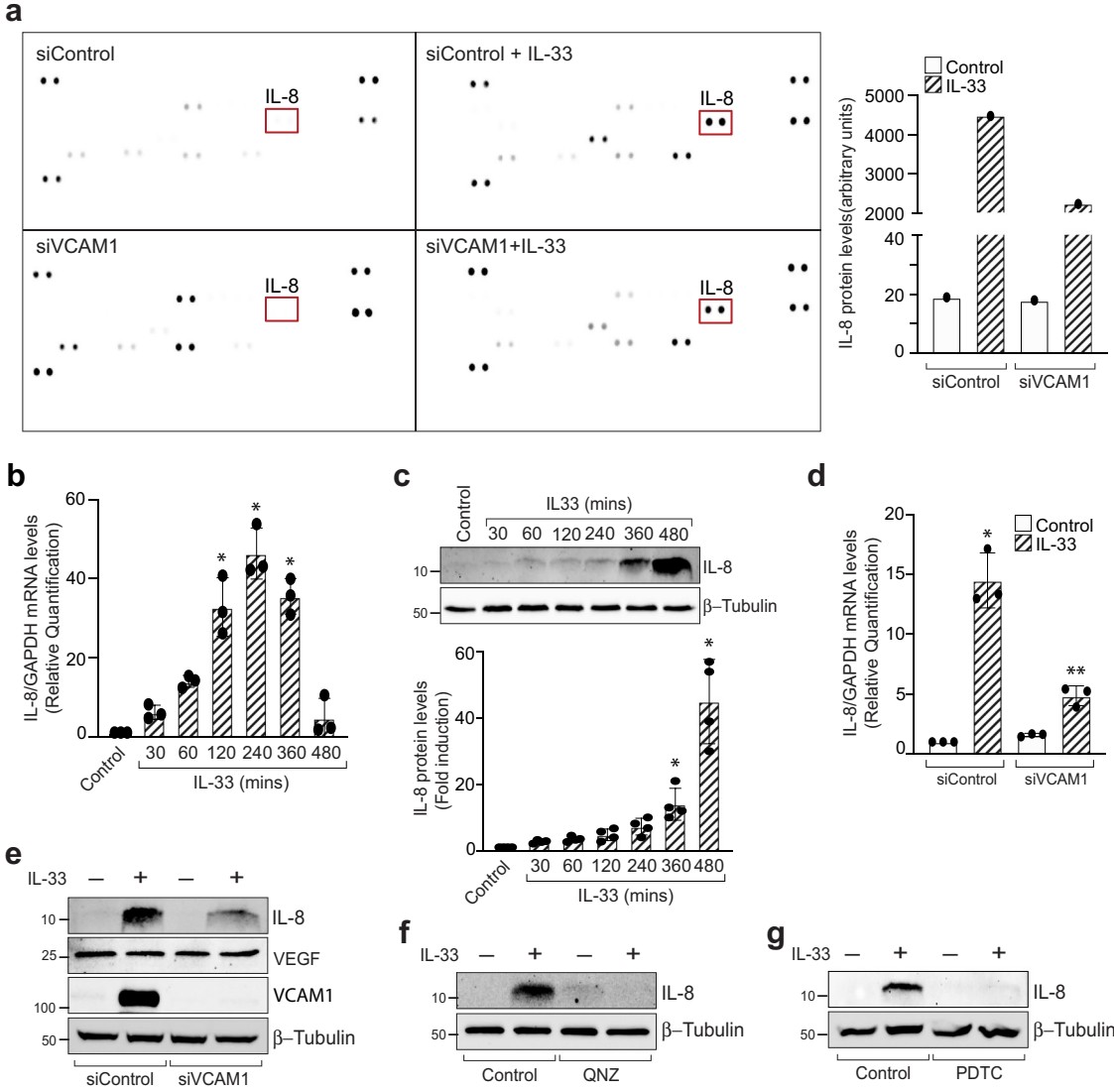

**Fig. 3 VCAM-1 regulates IL-33-induced IL-8 expression in HRMVECs. a** HRMVECs were transfected with control siRNA (siControl) or VCAM-1 siRNA (siVCAM-1), quiesced, treated with or without IL-33 (20 ng/mL) for 8 h, cell extracts were prepared and equal amount of protein from the mentioned groups were analysed for angiogenic genes using proteome profiler human angiogenesis array kit. **b**, **c** Quiescent HRMVECs were treated with IL-33 (20 ng/mL) for the indicated time periods and total cellular RNA and cell extracts were prepared and analysed by QRT-PCR and western blotting, respectively for IL-8 expression. **d**, **e** HRMVECs were transfected with control siRNA (siControl) or VCAM-1 siRNA (siVCAM-1), quiesced, total cellular RNA and cell extracts were prepared and analysed by QRT-PCR for IL-8 and by western blotting for the indicated proteins. **f**, **g** Quiescent HRMVECs were first treated with QNZ (10 μM, NFκB inhibitor) or PDTC (10 μM, NFκB inhibitor) for 30 min, followed with or without IL-33 (20 ng/mL) treatment for 8 h. The cell lysates were prepared and analysed for IL-8 levels by Western blotting and normalized to β-tubulin. $n = 3$ (**b**, **d**) or $n = 4$ (**c**) biologically independent samples per group, expressed as Mean ± SD. *$P < 0.05$ vs control or siControl, **$p < 0.05$ vs siControl + IL-33.

IL-8 expression in HRMVECs (Fig. 3f & g). These findings also signify the importance of NF-κB-VCAM-1 signaling in IL-33-induced IL-8 expression in HRMVECs.

IL-8 is an emerging regulator of angiogenesis and an elevated levels of IL-8 were observed in the vitreous of proliferative diabetic retinopathy patients[30]. To understand the functional significance of IL-33-induced IL-8 expression in HRMVECs, we evaluated the effect of IL-8 depletion on IL-33-induced migration, tube formation, and sprouting of HRMEVCs. We observed a significant reduction in IL-33-induced cell migration, wound healing, tube length, and the number of sprouts per bead in IL-8 siRNA transfected cells compared to control siRNA groups (Fig. 4a–e). The VEGF expression is unaffected by IL-8 depletion in HRMVECs (Fig. 4a).

**VCAM-1 via JunB regulates IL-8 promoter activity and expression in HRMVECs.** Soluble VCAM-1 is suggested to be a mediator of angiogenesis with few recent reports underlying its importance as a marker of angiogenesis in breast cancer[16,31]. Till now, VCAM-1 is shown to regulate endothelial-leukocyte inter-actions, and very little is known regarding the role of VCAM-1 in pathological angiogenesis. To understand how IL-33-induced VCAM-1 regulates IL-8 expression, we carried out the sequence analysis of the full-length human IL-8 promoter and found that it has one putative activator protein 1 (AP-1) binding motif at −221 nt (Fig. 5a). The IL-8 promoter region consisting of −342 to +59 promoter sequences were subcloned into pGL3 basic vector, HRMVECs were transfected with clones, treated with and without IL-33 for 6 h and luciferase activity was measured. The

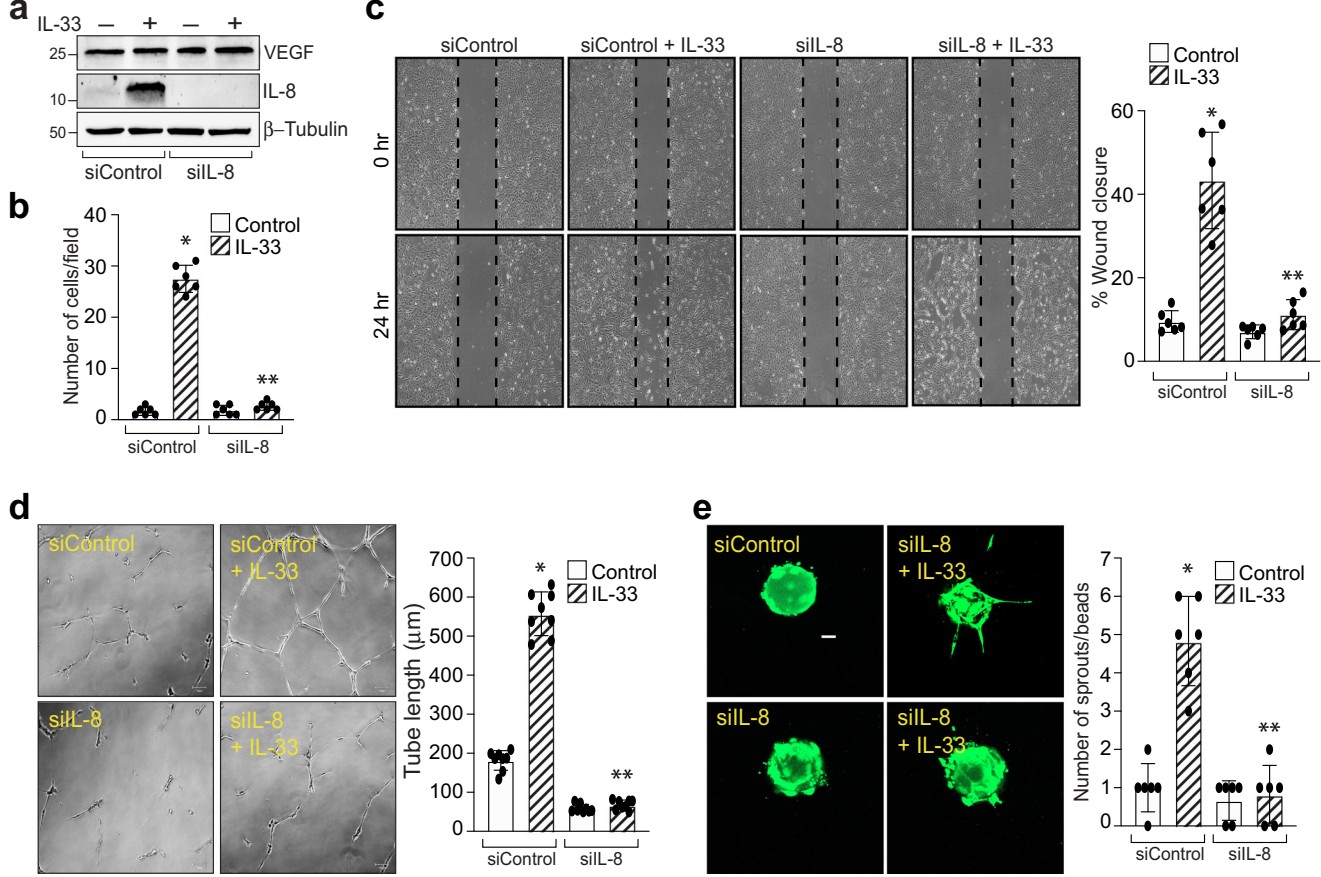

**Fig. 4 IL-8 mediates IL-33-induced angiogenic events in HRMVECs. a** HRMVECs were transfected with control siRNA (siControl) or IL-8 siRNA (siIL-8), after 48 h cell extracts were prepared and analysed for VEGF, and IL-8 levels by Western blotting and normalized to β-tubulin. **b–e** All the condition were same as in panel a, except that the cells were analyzed for IL-33 (20 ng/mL)-induced migration (**b**), wound healing (**c**), tube formation (**d**), and sprouting assay (**e**). n = 6 (**b, c, d, e**) biologically independent samples per group, expressed as Mean ± SD. *P < 0.05 vs siControl, **p < 0.05 vs siControl + IL-33. Scale bar represents 50 μm in panel **e**.

IL-33 treatment resulted in a 5-fold increase in the luciferase activity with pGL3-hIL-8 (0.44 kb) promoter construct (Fig. 5b), suggesting that AP-1 element at −221 nt is sufficient for IL-33-induced IL-8 promoter activity. To determine the AP-1 components involved in IL-33-induced IL-8 expression, we next studied the time course effects of IL-33 on the expression of various AP-1 subunits (Fos and Jun families of proto-oncogenes) in HRMVECs. As shown in Fig. 5c, exposure of HRMVECs to IL-33 significantly induces the expression of JunB with little or no effect on other Fos and Jun family of proto-oncogenes. We next looked for the role of VCAM-1 on IL-33-induced JunB expression in HRMVECs. The depletion of VCAM-1 level by its siRNA, resulted in a significant reduction in the expression of JunB in HRMVECs (Fig. 5d). JunB is a member of the AP-1 transcriptional factor and is shown to be involved in the formation of fetomaternal circulatory system[32]. Therefore, we next looked for the role of JunB on IL-33-induced IL-8 promoter activity and expression. The siRNA-mediated downregulation of JunB levels attenuated IL-33-induced IL-8 promoter activity and expression, but did not affect VEGF expression in HRMVECs (Fig. 5e & f).

To test whether IL-33-induces VCAM-1-JunB signaling in a murine model of oxygen-induced retinopathy (OIR), we studied the effect of OIR (relative hypoxia) on JunB expression in the retina. We observed an induced expression of JunB in the hypoxic retina (Fig. 5g) and genetic deletion of IL-33 attenuated hypoxia-induced JunB expression in the retina (Fig. 5h). We next tested the role of JunB on IL-33-induced angiogenic events in

HRMVECs. siRNA mediated JunB depletion attenuated IL-33-induced HRMVEC migration, wound healing, tube length, and sprouting (Fig. 6a–e).

**VCAM-1 regulate OIR-induced CXCL1 expression and pathological retinal neovascularization in murine retina**. Our observations suggested a role for VCAM-1 in IL-33-induced IL-8 expression in HRMVECs. The rodents do not express a direct homologs of IL-8, but CXCL1, CXCL2, and CXCL5-6 are considered functional homologs of IL-8 in rodents[33]. Therefore, we performed mRNA sequencing of retinal samples from normoxia and 3 days post hypoxia/OIR to understand the role of these functional homologs in pathological retinal neovascularization. All the genes which were significantly up-regulated or down-regulated in response to oxygen-induced retinopathy were represented as a heat map (Fig. 7a–c). We observed an increased expression of CXCL1 in the hypoxic retina as compared to normoxia (Fig. 7b & c). To further validate the RNA sequencing data, we studied the time-course effect of hypoxia/OIR on CXCL1 expression in the retina and observed a time-dependent induction in CXCL1 expression in the hypoxic retina, which was maximum at 72-hour post hypoxia and then starts to go down (Fig. 7d).

VCAM-1 knockout mice are embryonically lethal due to defects in placental development. Therefore, we used intravitreal injection of VCAM-1 siRNA to assess the importance of VCAM-1 deletion on OIR-induced pathological retinal angiogenesis in mice. We intravitreally injected 0.5 μl of 5% glucose solution

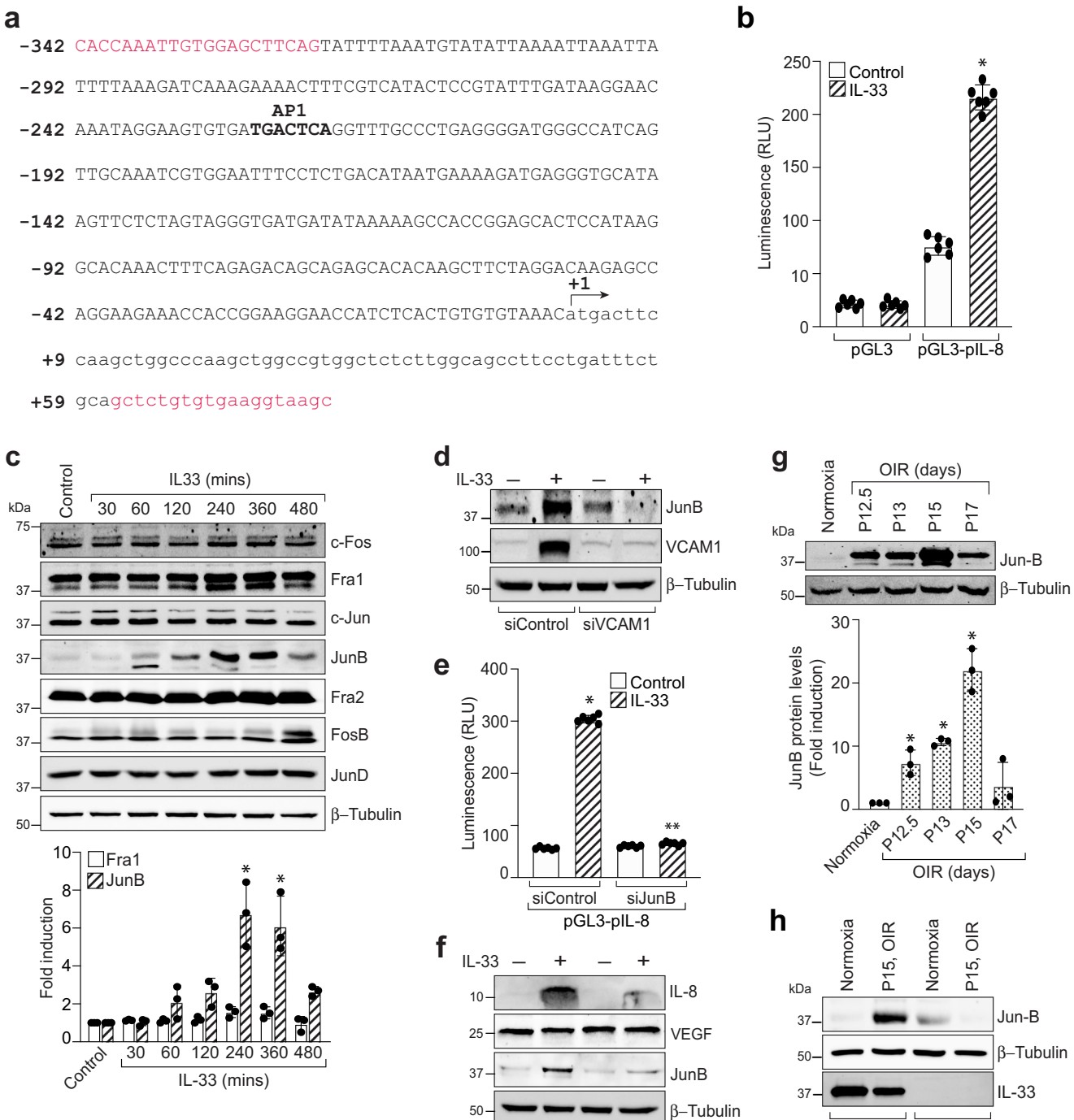

**Fig. 5 JunB regulates IL-33-induced IL-8 promoter activity. a** IL-8 promoter encompassing from −342 to +59 nt (401 kb) was cloned into pGL3-basic vector and sequenced. **b** HRMVECs were transfected with pGL3-basic vector or pGL3-IL-8 promoter (401 kb), growth-arrested, and treated with or without IL-33 (20 ng/mL) for 6 h, and the luciferase activities were measured. **c** Quiescent HRMVECs were treated with or without IL-33 (20 ng/mL) for various time periods and analysed by Western blotting for indicated proteins and normalized to β-tubulin. **d** HRMVECs were transfected with control siRNA (siControl) or VCAM-1 siRNA (siVCAM-1), quiesced, treated with or without IL-33 (20 ng/mL) for 6 h, cell extracts were prepared and analysed by Western blotting for JunB levels, and the blot was reprobed for VCAM-1 and β-tubulin to show the effects of siRNA on its target and off target molecules. **e** HRMVECs were transfected with pGL3-basic vector or pGL3-IL-8 promoter (1.215 kb) in combination with control siRNA (siControl) or JunB siRNA (siJunB), growth-arrested, and treated with and without IL-33 (20 ng/mL) for 6 h, and the luciferase activities were measured. **f** HRMVECs were transfected with control siRNA (siControl) or JunB siRNA (siJunB), quiesced, treated with or without IL-33 (20 ng/mL) for 8 h, cell extracts were prepared and analysed by Western blotting for the indicated proteins. **g** C57BL/6 mice pups were exposed to 75% oxygen and at various time periods of relative hypoxia, eyes were enucleated, retinas isolated, tissue extracts made and analysed by Western blotting for JunB levels using their specific antibodies and normalized to β-tubulin. **h** At P15, the retinal tissue extracts from IL-33$^{+/+}$ and IL-33$^{-/-}$ mice pups subjected to normoxia and hypoxia were analyzed for indicated proteins by Western blotting. $n = 6$ (**b, e**) or $n = 3$ (**c, g**) biologically independent samples per group, expressed as Mean ± SD. *$P < 0.05$ vs normoxia or control or siControl, **$p < 0.05$ vs siControl + IL-33.

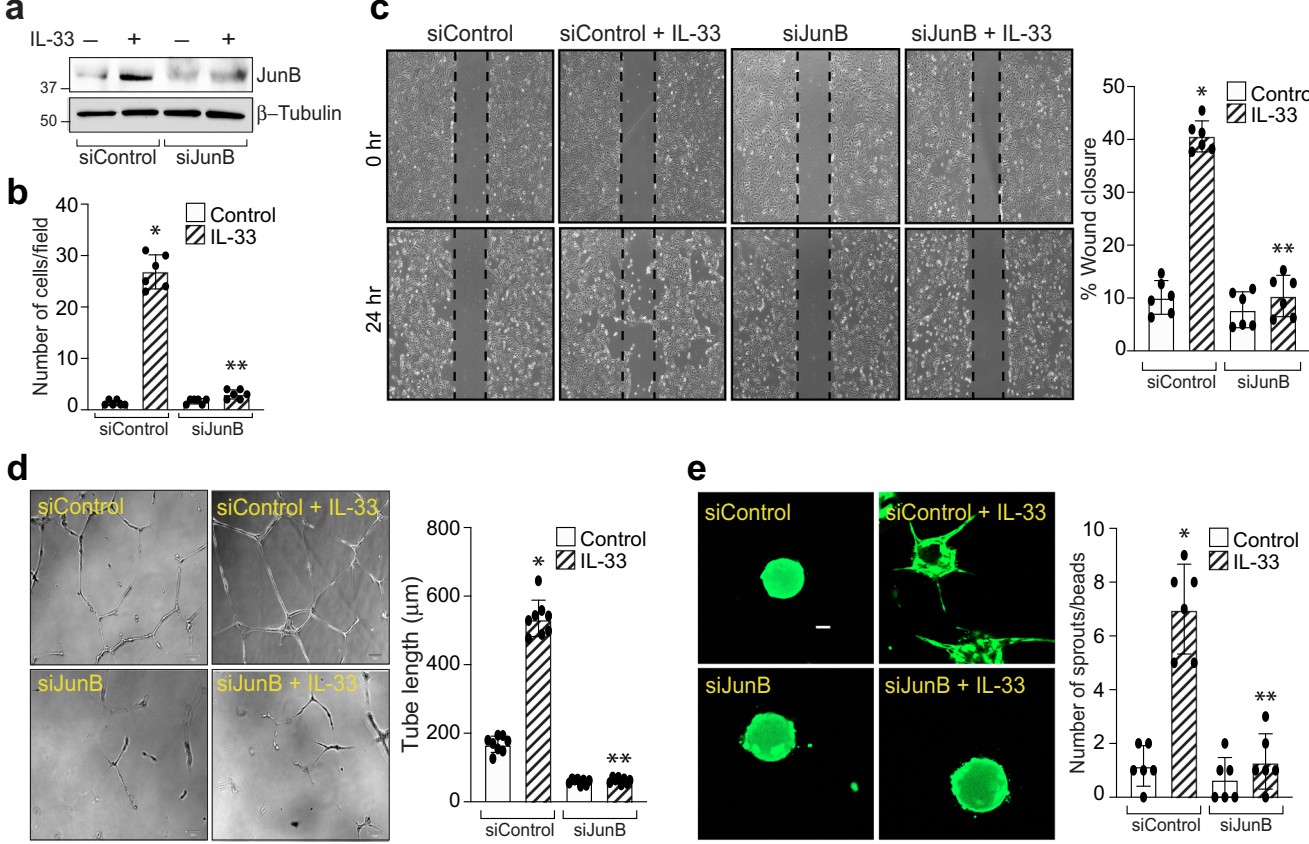

**Fig. 6 JunB mediates IL-33-induced angiogenic events in HRMVECs. a** HRMVECs were transfected with control siRNA (siControl) or JunB siRNA (siJunB), after 48 h cell extracts were prepared and analysed for JunB levels by Western blotting and normalized to β-tubulin. **b-e** All the condition are same as in panel (**a**), except that the cells were analyzed for IL-33 (20 ng/mL)-induced migration (**b**), wound healing (**c**), tube formation (**d**), and sprouting assay (**e**). $n = 6$ (**b, c, d, e**) biologically independent samples per group, expressed as Mean ± SD. *$P < 0.05$ vs siControl, **$p < 0.05$ vs siControl + IL-33. Scale bar represents 50 μm in panel (**e**).

containing in vivo-jetPEI® transfection reagent with 1 μg of control or VCAM-1 siRNA at P12 and P14. At P15, eyes were enucleated, retinas isolated, and retinal extracts were made and analyzed for CXCL1, JunB, and VCAM-1 level in normoxic and hypoxic retinas. The hypoxia-induced expression of CXCL1, JunB, and VCAM-1 was attenuated in mice injected with VCAM-1 siRNA compared to control siRNA group (Fig. 8a). At P17, the retinas were also stained with isolectin B4, flat-mounted, and quantified for the percentage of neovascularization or avascular area as detailed in "Methods" section. Downregulation in VCAM-1 level attenuated hypoxia-induced retinal neovascularization (Fig. 8b & c), and retinal endothelial cell (EC) tip cell formation (Fig. 8e). No significant difference were observed in terms of avascular area in control or VCAM-1 siRNA group (Fig. 8d). These observations suggest that the VCAM-1 regulates JunB-CXCL1 signaling and pathological retinal neovascularization in a murine model of OIR. A schematic diagram depicting the IL-33-NF-κB-VCAM-1-JunB signaling on IL-8/CXCL1 expression, and retinal neovascularization is presented in Fig. 9.

## Discussion

Retinal angiogenesis or neovascularization is the most common microvascular complication in various proliferative retinopathies. Retinal neovascularization is an underlying cause of vision impairment and legal blindness. The current treatment strategies against retinal neovascularization are effective in halting the progression of the diseases but is unable to restore the normal acuity[1]. Hence, a better understanding of the pathophysiology of

neovascularization is required to assess the effective targets. In retinal neovascular diseases, there is a crosstalk between immune cells and proangiogenic factors such as cytokines (TNFα and IL-1), adhesion molecules (VCAM-1 and ICAM-1), and growth factors (VEGF) that regulates angiogenesis[9].

IL-33 is a ligand of the ST2 receptor which is mainly expressed in endothelial and epithelial cells[34]. IL-33 is considered a stress-regulated cytokine that is associated with tissue damage[35]. IL-33 enhances the expression of adhesion molecules and cytokines and is involved in the modulation of inflammation and angiogenesis[36,37]. Considering the contrasting reports of the proangiogenic and antiangiogenic role of IL-33, we primarily evaluated the effect of IL-33-induced signaling on pathological retinal neovascularization. A recent report has shown that the expression of VCAM-1 and ICAM-1 is induced in the retinas of diabetic rats, suggesting a role for these adhesion molecules in diabetic retinopathy[38]. Therefore, we evaluated the role of endothelial cell adhesion molecules in a mouse model of OIR. In our studies, we observed an induced expression of VCAM-1 in the hypoxic retina of mice exposed to OIR, suggesting the importance of VCAM-1 in proliferative retinopathies. However, no significant changes were observed in the ICAM1 or E-selectin levels in the hypoxic retina. Our observations suggest that IL-33 induces VCAM-1 levels in HRMVECs and genetic deletion of IL-33 leads to attenuation in hypoxia/OIR-induced VCAM-1 level in retina. In concordance with our study, Demyanets et al.[24] have also shown that IL-33 induces VCAM-1 and ICAM-1 level in human coronary artery endothelial cells and suggested a role for

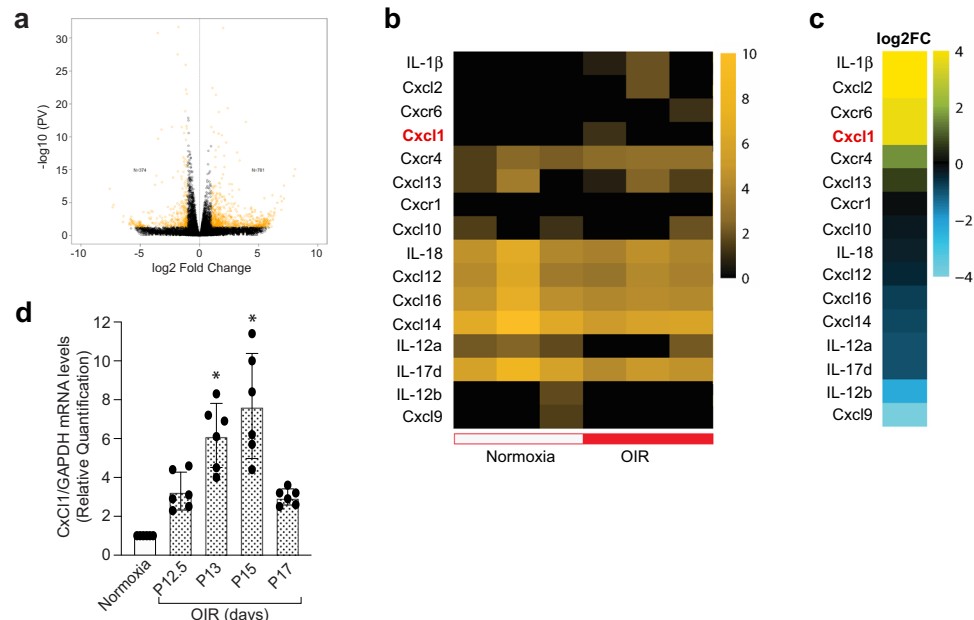

**Fig. 7 mRNA sequencing of retinal samples shows increased expression of CXCL1 (IL-8 functional homolog) in the hypoxic retina.** The C57BL/6 mice pups were exposed to 75% oxygen from P7 to P12 and then returned to room air. At P15, eyes from normoxia and 3 days post OIR/hypoxia were enucleated, retinas isolated, and total cellular RNAs were extracted and subjected to mRNA sequencing. **a** Volcano plot shows the number of genes (represented as yellow dots) that are significantly upregulated or downregulated in the hypoxic retina compared to normoxia. **b** Heatmap showing the expression profile of CXCL and its related genes in normoxia and hypoxia. **c** The genes that were significantly upregulated or downregulated in hypoxia compared to normoxia are presented as fold change. **d** At various periods of relative hypoxia, eyes were enucleated from C57BL/6 mice, retinas were isolated, and total RNA was extracted and analyzed for CXCL1 and GAPDH mRNA levels by QRT-PCR. $n = 6$ (**d**) biologically independent samples per group, expressed as Mean ± SD. *$P < 0.05$ vs normoxia.

IL-33 induced VCAM-1 and ICAM1 levels in atherosclerotic plaque development[24]. In addition, we also observed a role of IL-33 in hypoxia/OIR-induced retinal neovascularization, as genetic deletion of IL-33 in C57BL/6 mice resulted in reduced endothelial cell sprouting and neovascularization. We also looked for the role of VCAM-1 on IL-33-induced angiogenic events in HRMVECs and observed that siRNA-mediated downregulation of VCAM-1 level attenuated IL-33-induced endothelial cell sprouting, migration, wound healing, and tube formation. VEGF is a significant factor in OIR-induced retinal neovascularization[25], and a report indicates that VCAM-1 regulates VEGF-induced leukocyte recruitment to the ischemic retina[26]. Our findings suggest that IL-33 does not influence OIR-induced VEGF expression, and the VCAM-1-JunB-IL-8 signaling does not affect VEGF expression in HRMVECs.

Many studies including us have shown that IL-33 regulates NF-κB signaling pathway in endothelial cells[21], and it has also been demonstrated that murine and human VCAM-1 promoter contain NF-κB DNA binding sequences[39]. We observed a significant augmentation in the expression of NF-κB signaling molecules and pharmacological inhibition of NF-κB signaling resulted in attenuation of IL-33-induced VCAM-1 expression in HRMVECs. Taken together, our findings suggest a decisive role of NF-κB signaling on IL-33-induced VCAM-1 expression both in human retinal endothelial cells and murine hypoxic retina.

In recent years, increasing amounts of evidence have highlighted the importance of VCAM-1 in angiogenesis. It has been demonstrated that siRNA mediated down regulation or antibody specific inhibition of VCAM-1 not only inhibits TNFα-induced human umbilical vein endothelial cell migration and tube formation, but also rat aortic vessel sprouting[40]. There are enough reports which suggest a role for VCAM-1 in angiogenesis, but how VCAM-1 regulate these angiogenic events in endothelial cells is still unknown. In this regard, we performed a protein profiler human angiogenesis array in HRMVECs and found that IL-8 expression was upregulated with IL-33 treatment, and downregulation of VCAM-1 levels using its siRNA attenuated IL-8 expression. We confirmed our protein profiler human angiogenesis array results using QRT-PCR and western blotting techniques and observed that IL-33 induces IL-8 expression both at mRNA and protein levels. In addition, downregulation of VCAM-1 levels using its siRNA significantly blocked IL-33-induced IL-8 levels, both at mRNA and protein levels. Similarly, pharmacological inhibition of NF-κB signaling, downregulates IL-33-induced IL-8 expression in HRMVECs. Interleukin-8 (CXCL8) is a proinflammatory chemokine, which is shown to be involved in the progression of various cancers[41]. Additionally, increased levels of IL-8 were reported in proliferative diabetic retinopathy and proliferative vitreous retinopathy patients[42]. To ascertain the functional role of IL-8 in IL-33-induced angiogenic events in HRMVECs, migration, tube formation, and sprouting assays were performed in control and IL-8 siRNAs transfected HRMVECs. These assays revealed a significant reduction in the percentage of wound healing, tube length, invasion, and the number of sprouts per bead in IL-8 siRNA transfected HRMVECs. All these observations signify the importance of IL-33-induced VCAM-1 on IL-8 expression and angiogenesis.

Accumulating evidence in the literature demonstrated that the JNK pathway that regulates AP-1 transcription activity plays a critical role in the biological actions of IL-33[43]. Therefore, the underlying mediator in IL-8 regulation was evaluated by sequence analysis of the full-length IL-8 promoter and we observed that the AP-1 binding site at −221 nt proximal to the transcription start site is essential for IL-33-induced IL-8 promoter activity. AP1 transcription factor is composed of Jun and Fos families of proteins and therefore we looked for the role of these proteins on

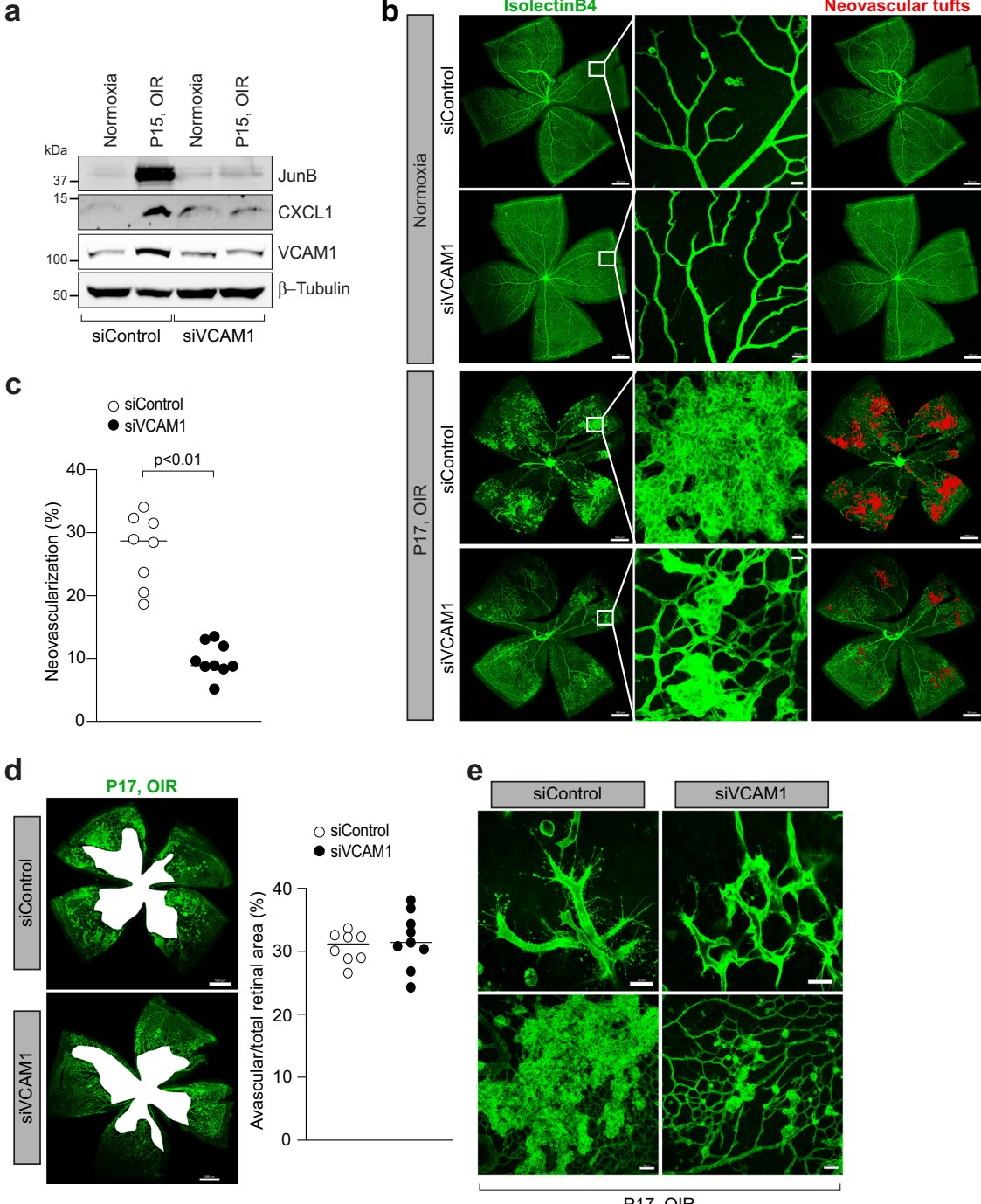

**Fig. 8 VCAM-1 regulates OIR-induced retinal neovascularization. a** C57BL/6 mice pups were exposed to 75% oxygen (P7-P12), returned to room air and administered intravitreally with 0.5 µl of 5% glucose solution containing in vivo-jetPEI® transfectoin reagent with 1 µg of control or VCAM-1 siRNA at P12 and P14. At P15, the eyes were enucleated, retinas isolated, tissue extracts were prepared and analysed for JunB, and CXCL1 levels by Western blotting and the blot was reprobed for VCAM-1 and β-tubulin to show the effects of siRNA on its target and off target molecules. **b**, **d**, **e** Everything is same as in panel (**a**), except that at P17, the eyes were enucleated, retinas isolated, stained with isolectin B4, flat mounts prepared and examined for retinal neovascularization (**b**), avascular area (**d**), and endothelial tip cell formation (**e**). The middle column in panel b shows the higher magnification of the area selected. Neovascularization is highlighted in red in the third column of panel (**b**). **c**, **d** The bar graphs represent quantitative analysis of neovascularization (**c**) and avascular area (**d**). *n* = 8 (**c, d**) biologically independent eyes per group, and expressed as Mean ± SD. *P < 0.05 vs siControl. Scale bar represents 500 µm in panel (**b**) left column and right column, 20 µm in panel (**b**) middle column, 500 µm in panel (**d**), 20 µm in panel (**e**) upper row, 50 µm in panel (**e**) lower row.

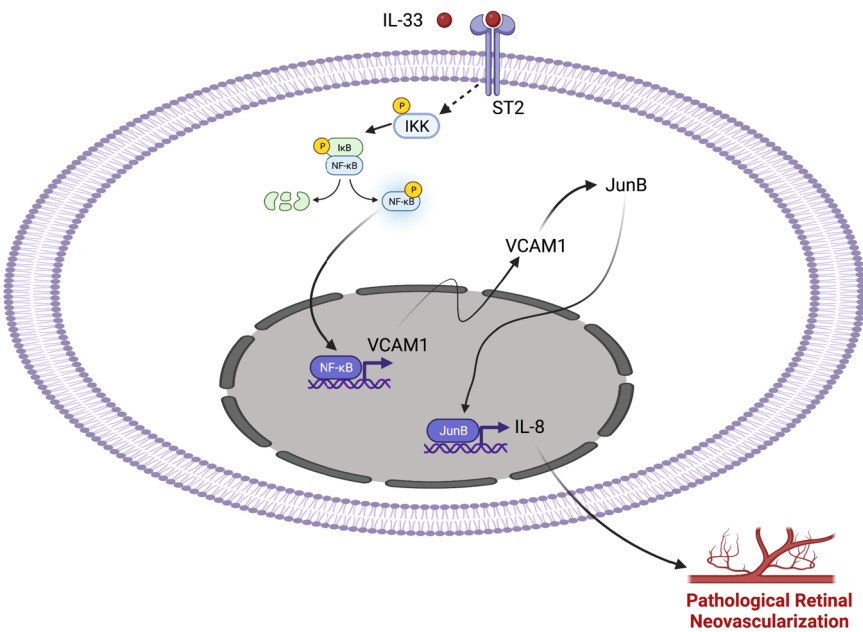

**Fig. 9 IL-33-NF-κB-VCAM-1-JunB signaling regulates IL-8 expression and pathological retinal angiogenesis.** Schematic diagram showing the role of IL-33 induced NF-kB-VCAM-1-JunB-IL-8/CXCL1 signaling on pathological retinal neovascularization. Figure 9 was created with BioRender.com.

IL-33-induced IL-8 promoter activity. We observed that IL-33 significantly induced the expression of JunB, with little or no effect on other Jun and Fos family members. Studies have shown that JunB homodimers induce cell proliferation, while its heterodimerization with cJun reduces cell proliferation[44]. Our observations support this possibility, as IL-33 only induces JunB levels in HRMVECs with little or no effect on the expression of other family members of AP1 transcriptional factors. Various studies have outlined the positive role of JunB in tumor invasion and progression[45–47]. Consistent with these observations, we also observed that siRNA-mediated downregulation of JunB significantly reduced IL-33-induced IL-8 promoter activity and expression. In addition, JunB depletion resulted in reduced retinal endothelial cell sprouting, migration, wound healing, and tube formation.

IL-8 is a well-characterized pro-angiogenic chemokine and is shown to be associated with tumor angiogenesis[48]. High concentrations of IL-8 were observed in vitreous of proliferative diabetic retinopathy patients, suggesting the importance of IL-8 in proliferative retinopathies[30]. A direct homolog of IL-8 is absent in rodents but CXCL1, CXCL2, and CXCL5-6 are considered as functional homologs of IL-8 in rodents and are reported to participate in the disease pathology of various animal models[49–51]. As we observed an induced expression of IL-8 by IL-33 in HRMVECs, we looked for the expression of various CXCL proteins in the murine model of oxygen-induced retinopathy using RNA sequencing. The mRNA-sequencing results showed an induced expression of CXCL1 in the hypoxic retina compared to normoxia, which was further verified by QRT-PCR. CXCL1 is a functional homolog of IL-8 in rodents and studies have shown that CXCL1 have therapeutic efficacy in limiting inflammation[52].

VCAM-1 is an endothelial cell adhesion molecule, which is shown to be involved in inflammation and cancer. It was also reported that VCAM-1 regulates the proangiogenic effect of oxidative stress in ischemic retina[23]. Despite these reports, nothing is known regarding the direct role of VCAM-1 on OIR-induced pathological angiogenesis. In the present manuscript, we have shown that VCAM-1 via JunB mediated IL-8 expression regulates IL-33-induced angiogenic events in human retinal endothelial cells. Therefore, we next looked for the direct role of

VCAM-1 on pathological retinal neovascularization in a murine model of oxygen-induced retinopathy. As VCAM-1 knockdown is embryonically lethal due to placental defect, we used intravitreal injections of VCAM-1 siRNA to deplete VCAM-1 level in the retina and studied its effect on OIR-induced pathological retinal neovascularization. Downregulation of VCAM-1 level attenuated hypoxia induced JunB and CXCL1 levels in the hypoxic retina, which further resulted in reduced endothelial cell sprouting and pathological neovascularization. No significant changes were observed in the avascular retinal area between the two groups. These findings suggest that VCAM-1 has a significant role in pathological retinal neovascularization with no significant effect on normal retinal repair.

In conclusion, our findings provide evidence of an underlying mechanism in the pathophysiology of retinal angiogenesis and identify potential therapeutic targets. Our findings demonstrate that VCAM-1 via JunB-IL-8 signaling regulates sprouting angiogenesis in human retinal endothelial cells. In addition, VCAM-1 via JunB-CXCL1 signaling regulates vessel anastomosis and tufts formation in mouse model of oxygen-induced retinopathy. The IL-33-NF-κB-VCAM-1-JunB-IL-8/CXCL1 signaling unravels the advanced strategies in the amelioration of proliferative retinopathies.

## Methods

**Reagents**. VCAM-1 Mouse specific (39036, dilution 1:1000), anti-P-NF-κB (3033, dilution 1:1000), anti-NF-κB (8242, dilution 1:1000), anti-P-IKKα/β (2697, dilution 1:1000), anti-IKKβ (8943, dilution 1:1000), anti-P-IκBα (2859, dilution 1:1000), anti-IκBα (4814, dilution 1:1000), β-tubulin (2128, dilution 1:1000), FosB (2251, dilution 1:1000), Fra2 (19967, dilution 1:1000), JunD (5000, dilution 1:1000) antibodies were obtained from Cell Signaling Technology (Beverly, MA). Anti-VEGF (sc-57496, dilution 1:500), anti-VCAM-1 (sc-13160, dilution 1:500), anti-ICAM1 (sc-8439, dilution 1:500), anti-JunB (sc-8051, dilution 1:500), anti-Fra1 (sc-28310, dilution 1:500), anti-Fos/c-Fos (sc-166940, dilution 1:500), and anti-c-Jun (sc-74543, dilution 1:500) antibodies were purchased from Sant Cruz Biotechnology (Dallas, Texas). Recombinant anti-IL-8 (ab235584, dilution 1:1000) and anti-CXCL1 (ab86436, dilution 1:1000) antibodies were purchased from Abcam (Cambridge, CA). Growth factor reduced Matrigel (354230) was purchased from BD Biosciences (Bedford, MA). Recombinant Human IL-33 protein (3625-IL-010/CF), goat anti-IL-33 antibody (AF3626), and protein profiler human angiogenesis array kit (ARY007) were purchased from R&D systems (Minneapolis, MN). EGM2 medium was purchased from Lonza (Basel, Switzerland). The small interfering RNAs (siRNAs) for human VCAM-1 [s14759, sense sequence (5'- > 3') GGAGUUAAUUUGAUUGGGAtt, Antisense

sequence (5'->3') UCCCAAUCAAAUUAACUCCtt], human JunB [s7661, sense sequence (5'->3') CUCUCUACACGACUACAAAtt, Antisense sequence (5'->3') UUUGUAGUCGUGUAGAGAGag], human IL-8 [s7327, sense sequence (5'->3') GAACUUAGAUGUCAGUGCAtt, Antisense sequence (5'->3') UGCACUGACAU CUAAGUUCtt], mouse VCAM1 [s75916, sense sequence (5'->3') CCAUUGAAG AUACCGGUAAtt, antisense sequence (5'->3') UUCCCGGUAUCUUCAAUGGtg], and scrambled control (4390844) were purchased from Ambion (Carlsbad, CA). Trizol reagent (15596026) was also purchased from Ambion (Carlsbad, CA). Cytodex 3 microcarrier beads (C3275), aprotinin (A6279), thrombin (T8885) and 1 Bromo-3-chloropropane (B9673) were obtained from SIGMA-ALDRICH (St. Louis, MO). VECTASHIELD Antifade mounting medium with DAPI (H-1500), and without DAPI (H-1700) were purchased from Vector laboratories (Burlington, Ontario, Canada). Bovine fibrinogen (J63276) was purchased from Alfa Aesar (Tewksbury, MA). Cell Tracker Green (C7025), isolectin B4-594, were bought from Invitrogen (Carlsbad, CA). QNZ (SC-200675) was purchased from Santa Cruz Biotechnology (Dallas, Texas). Pyrrolidinedithiocarbamate ammonium (PDTC, cat:0727) was purchased from TOCRIS. Nano-Glo dual luciferase reporter assay system (N1610) was purchased from Promega. Taq Man universal master mix ll, no UNG (4440040) and high-capacity cDNA Reverse transcription kit were purchased from Applied Biosystems, Foster City, CA. The main table in the reporting summary includes all oligo sequences, cell line sources, antibodies used, and their concentrations.

*Experimental animals.* We obtained C57BL/6 mice from Charles River Laboratories (Wilmington, MA). IL-33[flox/flox] mice (030619), and E2a-Cre mice (003724) were from Jackson Laboratory (Bar Harbor, ME). The mice were housed and bred in a 12-hour light/12-hour dark cycle environment and fed ad libitum water and food. Both male and female mice pups (of age P12 to P17) were used for this study. The institutional Animal Care and Use Committee of Wayne State University, Detroit, MI approved all the animal experiments.

**IL-33 knockout mice**. We generated IL-33 knockout mice by crossbreeding IL-33[flox/flox] mice[53] with E2a-Cre mice. The germline deletion of IL-33 in mice occurs due to E2a-Cre recombinase[54].

**Cell culture**. The HRMVECs were purchased from Applied Cell Biology Research Institute (ACBRI 181, Kirkland, WA). The cells were grown in an EGM2 medium containing 10 μg/ml gentamycin, and 0.25 μg/mL amphotericin B and maintained at 37 °C in a humidified 95% air and 5% $CO_2$ atmosphere. The cells in the passage range between 5 and 10 were used for experiments. Prior to experimentation, HRMVECs were synchronised in serum-free medium for 24 h (h) to attain quiescence.

**Cell migration**. The modified Boyden chamber method was used to perform cell migration[55]. Briefly, the cells with and without transfection with indicated siRNA were quiesced, and then plated at a concentration of $5 \times 10^4$ cells per insert in the upper side of a matrigel-coated-8-μm cell culture insert. Vehicle or IL-33 was added to the upper chamber to assess the effect of indicated siRNA on IL-33-induced HRMVECs migration. This led to the segregation of migrated cells on the lower surface and non-migrated cells in the upper side of the membrane. Migrated cells were fixed in methanol for 10 min and non-migrated cells were removed from the upper side by cotton swabs. The membrane was then stained with DAPI and observed under EVOS M5000 microscope (ThermoFisher Scientific, Waltham, MA). The migrated cells were presented as the number of migrated cells per field.

**Wound healing assay**. Cells were grown in a 6 well culture dishes, quiesced in serum free EBM2 media for 24 h at 37 °C. In each well, with the help of a sterile plastic micropipette tip a straight-edged wound was created with cell-free zone. The cells were washed and treated with or without IL-33 (20 ng/mL) in a serum free media containing 5 mM hydroxyurea for 24 h. The cell migration was observed using EVOS M5000 microscope (Thermo Fischer Scientific, Waltham, MA). The data were analyzed using ImageJ version 1.43 software. The cell migration was expressed as percent wound healing [total wound area (0 h) - area (after 24 h/total wound area × 100)].

**Sprouting assay**. For HRMVECs sprouting assay, three-dimensional sprouting assay was carried out[56]. Firstly, cells were transfected with the control and targeted siRNAs. The cells were with microcarrier beads (Cytodex 3) for attachment at 37 °C for overnight. The beads were then embedded in a 3D fibrin gel, fibroblast cells were seeded above the fibrin gel. The cell sprouting was observed using Zeiss LSM 800 microscope and image analysis software Zen was used to capture sprout images.

**Tube formation**. For tube formation assay, growth factor reduced Matrigel was used[57]. The HRMVECs were grown to full confluency and quiesced in serum free media for 24 h. Firstly, 24 well plate was coated with matrigel, and $1 \times 10^5$ cells were platted to the Matrigel coated plate. The cells were treated with or without IL-33 (20 ng/mL) for 6 h at 37 °C, and the tube formation was observed under EVOS M5000 microscope. The effect of siRNAs on IL-33-induced HRMVECs, tube

formation was evaluated by transfecting the cells with siRNA, quiescing, and measuring tube formation, as mentioned above. The tube length was calculated (in micrometer) using NIH ImageJ version 1.43.

**Western blotting**. The protein from HRMVECs or retinal extracts was resolved on polyacrylamide gel electrophoresis (SDS-PAGE). The proteins were electrophoretically transferred to a nitrocellulose membrane, blocked, and probed with primary and secondary antibodies. After washing, super signal west pico plus (an enhanced chemiluminescent detection) was used to detect the antigen-antibody complexes.

**Transfections**. HRMVECs were transfected with 100 nanomoles of control or target siRNA molecules using Lipofectamine 3000 transfection reagent according to the manufacturer's instructions.

**Intravitreal injections**. At P12 and P14, the pups were given 1 μg of the indicated siRNA in 0.5 μl of 5% glucose solution containing in vivo-jetPEI® transfection reagent intravitreally using a 35 G needle.

**Oxygen-induced retinopathy (OIR)**. At postnatal day 7 (P7), mice pups with dams were exposed to 75% oxygen (hyperoxia) for 5 days (P7 to P12), in Bio-Spherix chamber and on P12 mice pups along with dams returned to room air[58]. For control group, the mice pups of same age were kept at 21% oxygen (i.e., room air). The mice pups were euthanized at P17, eyes were enucleated and fixed. Retinas were isolated and staining was done with isolectin B4. Then flat mount of retinas was prepared and examined under a Zeiss LSM 800 confocal microscope. The percentage (%) of retinal neovascularization (fluorescence intensity from highlighted region/total fluorescence intensity × 100) was calculated using Nikon NIS-Elements Advanced research software[21]. The avascular area percentage (avascular/total retinal area × 100) was estimated using NIH ImageJ software.

**Quantitative real-time-PCR (QRT-PCR)**. The total cellular RNA was extracted from cells as well as tissue samples using Trizol. Following the manufacturer's protocol, cDNA was synthesized using the High-Capacity cDNA reverse transcription kit (Applied Biosystems, Foster City, CA). The TaqMan Gene Expression Assays for human CXCL-8 (Hs00174103_m1), human GAPDH (Hs02786624_g1), mouse CXCL-1 (Mm04207460_m1), and mouse GAPDH (Mm99999915_g1) were used for PCR amplification. PCR amplification was performed using 7300 Real-Time PCR Systems (Applied Biosystems) according to the manufacturer's cycling parameters, and Ct values were obtained using the 7300 real-time SDS version 1.4 program.

*Cloning of IL-8 promoter.* The IL-8 promoter region was PCR-amplified from genomic DNA using primers (forward: 5'- TACGGGGTACCCACCAAATTGTG GAGCTTCAG -3' and reverse: 5'- CGTCAGCTAGCGCTTACCTTCACACAGA GC-3'). The PCR product of ~422 bp was then digested with KpnI and NheI restriction enzymes and cloned into KpnI and NheI of the pGL3-basic vector (4818 bp, Promega) to yield pGL3-hIL-8 construct (5240 bp).

*Luciferase assay.* HRMVECs cells grown in a 6-well plate were transiently transfected with pGL3-basic vector (4818 bp) or pGL3-hIL-8 (5240 bp) promoter constructs using Lipofectamine 3000 transfection reagent. The cells were growth arrested overnight in serum-free media. Following, cells were treated with and without IL-33 (20 ng/mL) for 6 h. Then a volume of ONE-Glo™ EX Reagent (N1610, Promega) equal to the volume of culture medium was added to each well. The sample plate was incubated for at least 3–10 mins at RT or mixed on an orbital shaker (300–600 rpm). Then the samples were transferred to 94-well plate and firefly luminescence was measured using BIOTEK SYNERGY H1 microplate reader (with Gen5™ Data Analysis Software) and expressed as relative luciferase units (RLU).

**RNA sequencing**. At P15, the normoxic and OIR-exposed C57BL/6 mice pups were euthanized, retinas isolated, total cellular RNA was extracted by TRIzol reagent, and sent for mRNA sequencing. Gene expression was identified using 3'mRNA-seq libraries generated from Lexogen's Quant-seq kit before sequencing ($1 \times 75$ bp) on a NovaSeq 6000. Sequencing data was aligning to the mouse genome (mm10) using STAR before tabulating counts across genes (htseq-count). Differentially expressed genes were determined in R using edgeR with heatmaps generated using the pheatmap package. Significantly altered genes (|log fold change| = 2; *p*-value = 0.05) were selected for Gene Ontology analysis (RDA-VIDWebService). The mRNA sequencing data has been submitted to the Gene Expression Omnibus (code GSE209654) and is now available to the public.

**Statistics and reproducibility**. All the experiments were performed in triplicates and the data is presented as mean ± SD. To compare the difference between two groups two tailed *t* test was used. One-way ANOVA with Tukey's correction was used to assess the comparison in experiments having more than two groups. For

statistical analysis GraphPad Prism 9 (Prism) was used. All $P < 0.05$ values were statistically significant.

**Reporting summary**. Further information on research design is available in the Nature Portfolio Reporting Summary linked to this article.

## Data availability

The supplementary figures 1–8 contain all the uncropped Western blot images presented in the article. The statistical source data for all the graphs presented in research article are available as Supplementary Data 1 (Excel file), and the remaining datasets are available from the corresponding author on request. The mRNA sequencing data has been submitted to the Gene Expression Omnibus (code GSE209654) and is available to the public.

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

## Acknowledgements
The present research work was supported by National Institutes of Health grants (EY029709 to NKS). Supported by a Research to Prevent Blindness unrestricted grant to Kresge Eye Institute, and by P30EY04068 (LDH) at Wayne State University. The Schematic diagram of the present research is created with BioRender.com.

## Author contributions
G.K. performed proteome prolifer array, cell migration, sprouting, proliferation, tube formation, and Western blot analysis and wrote the manuscript. D.S. performed IL-8 promoter cloning and luciferase assay. S.B. performed intravitreal injections, OIR and Western blot analysis. C.S.M. provided valuable suggestions. K.G. performed 3'-mRNA sequencing analysis. N.K.S. performed intravitreal injections, OIR, designed the project, supervised the study, interpreted the data and wrote the manuscript.

## Competing interests
The authors declare no competing interests.
