## [Peer Review File · Communications Biology]

Reviewers' comments:

Reviewer #1 (Remarks to the Author):

The authors of this study investigated the expression of various vascular adhesion molecules in the murine model of oxygen-induced retinopathy (OIR), and observed that VCAM-1-JunB-IL-8 signaling plays a crucial role in retinal neovascularization, by utilizing various assays on IL-33 knockout and wildtype mice. This study demonstrated that VCAM-1 via JunB-IL-8 signaling regulates sprouting angiogenesis in human retinal endothelial cells. Comprehensive experiments were conducted to obtain the convincing evidence. Whereas the study is not so innovative as proteins like VCAM-1, NF- κ b were already investigated and known closely related with hypoxia induced neovascularization.

Reviewer #2 (Remarks to the Author):

The article titled "Vascular cell-adhesion molecule 1 (VCAM-1) regulates JunB-mediated IL-8/CXCL1 expression and pathological neovascularization" is well written.

My comments are as followed:

-Some ophthalmic terminologies should be standardized. eg. "wet macular degeneration" should be "wet age-related macular degeneration".

-The references should be attached to "The anti-VEGF therapies are effective but are often associated with macular edema, reduced night vision, tractional detachment, and atrophy" in the introduction.

-This study clarified the IL-33-NF- κ B-VCAM-1-JunB-IL-8/CXCL1 signaling pathway. The increasing expression of VCAM-1 in ischemic retina was confirmed by the animal experiment. However, it is more convincing to compare VCAM-1 differences between patients and healthy controls than to directly conduct animal experiments

Reviewer #3 (Remarks to the Author):

This study describes the important role of VCAM-1 in retinal neovascularization by in vivo and in vitro experiments, and explores the upstream and downstream regulatory mechanism of IL33/NF-KB/VCAM-1/JunB/IL8 in oxygen-induced retinopathy (OIR). The manuscript is not well-written and has many mistakes. Please take the following concerns into consideration.

1. To maintain the consistency and readability of the whole article, the time points in the OIR model should be postnatal (P) with numbers, rather than the ones displayed in Figure 1 (hypoxia hours).
2. Please indicate the reasons why choose P15 to observe the relative expression of each molecule in OIR retinas. How about at P17?
3. There is a mistake in labeling Fig. 2g, h, I, which seems to be different from the main text and figure legends.
4. The observation time point of pNF-KB in Fig. 2f is not consistent with VCAM-1 in Fig. 1c, please indicate the reasons.
5. In Fig. 2g, h, the processing time of IL33 was 6h, rather than 8h which has the largest increasing trend in Fig. 1c. Moreover, in Fig. 3, the authors chose 8h to process the cells. Please indicate the reasons.
6. CXCL6 is also the homolog of IL-8, and the increased foldchange in Fig. 7b is higher than CXCL1. Therefore, it is suggested to validate CXCL6 by QRT-PCR.
8. The study used intravitreal injection at P12 and P14. To inject twice seems to have strong experimental uncertainty which may affect the in vivo results in the OIR model.

9. It is also necessary to explore and discuss the relevance of VEGF and VCAM-1 in retinal neovascularization. For example, the authors may check the expressions of VEGFA in some experiments.

10. In Fig. 8b,c,d, it seems that the retinas were collected at P15 from figure legends, while the results part mentioned they used the results at P17, which is quite chaotic.

We are thankful to the reviewers for their suggestions. We have now incorporated the reviewers' suggestions in the revised manuscript. All the changes in the manuscript are marked as track changes.

Reviewer #1:

- 1) The authors of this study investigated the expression of various vascular adhesion molecules in the murine model of oxygen-induced retinopathy (OIR) and observed that VCAM-1-JunB-IL-8 signaling plays a crucial role in retinal neovascularization, by utilizing various assays on IL-33 knockout and wildtype mice. This study demonstrated that VCAM-1 via JunB-IL-8 signaling regulates sprouting angiogenesis in human retinal endothelial cells. Comprehensive experiments were conducted to obtain the convincing evidence. Whereas the study is not so innovative as proteins like VCAM-1, NF-kb were already investigated and known closely related with hypoxia induced neovascularization.

Answer: *We are thankful to the reviewer that he felt that comprehensive experiments were carried out in the present manuscript to gather compelling evidence.*

We agree with Reviewer #1 that there is a report which outlines the role of VCAM1 in oxidative stress-induced retinal neovascularization (PNAS, 2011,108(35), 14614-14619), but in the mentioned manuscript, the authors have shown that blockade of VCAM1 does not affect the retinal NV in ischemic retina (sod+/- mice), but it eliminates the oxidative stress-induced increase in retinal neovascularization in the ischemic retina (sod-/- mice). In contrast, we are the first to establish that IL-33-induced VCAM-1-JunB-IL-8/CXCL1 signaling governs retinal endothelial cell sprouting and angiogenesis in both human retinal microvascular endothelial cells and in a murine model (C57BL/6 mice) of oxygen-induced retinopathy. These are novel scientific findings, and no previous research has shown that VCAM1 governs pathological angiogenesis via JunB-IL-8/CXCL1 signaling.

Reviewer #2:

- 1) Some ophthalmic terminologies should be standardized. eg. "wet macular degeneration" should be "wet age-related macular degeneration".

Answer: *We have now corrected it in the revised manuscript (please refer to page 3, line 83).*

- 2) The references should be attached to "The anti-VEGF therapies are effective but are often associated with macular edema, reduced night vision, tractional detachment, and atrophy" in the introduction.

Answer: *We have now provided references for it in the revised manuscript (please refer to page 3, line 90 and references 4-6).*

- 3) This study clarified the IL-33-NF- κ B-VCAM-1-JunB-IL-8/CXCL1 signaling pathway. The increasing expression of VCAM-1 in ischemic retina was confirmed by the animal experiment. However, it is more convincing to compare VCAM-1 differences between patients and healthy controls than to directly conduct animal experiments.

Answer: *We are thankful to the reviewer for asking this question. A few studies have demonstrated elevated levels of adhesion molecules, notably VCAM1, in the vitreous humor of individuals with proliferative diabetic retinopathy and hypothesized that VCAM1 might play a role in pathological retinal neovascularization. We have now included it in the revised manuscript (please refer to pages 6, lines 204-208).*

Reviewer #3:

- 1) To maintain the consistency and readability of the whole article, the time points in the OIR model should be postnatal (P) with numbers, rather than the ones displayed in Figure 1 (hypoxia hours).

Answer: *We are thankful to the reviewer for pointing this out to us. We have now included it in the revised manuscript (Please refer to figure 1b & 1g; 2i, 5g & h, 7d, and 8a).*

- 2) Please indicate the reasons why choose P15 to observe the relative expression of each molecule in OIR retinas. How about at P17?

Answer: *We chose P15 to examine the relative expression of each molecule in OIR retinas because we saw a considerable increase in JunB, CXCL1, and VCAM1 expression at P15, followed by a decrease at P17 (Please refer to figure 1b, 5g, and 7d).*

- 3) There is a mistake in labeling Fig. 2g, h, i, which seems to be different from the main text and figure legends.

Answer: *We are sorry for this error, and we have now corrected it in the revised manuscript (please refer to Page 6, lines 228-230).*

- 4) The observation time point of pNF- κ B in Fig. 2f is not consistent with VCAM-1 in Fig. 1c, please indicate the reasons.

Answer: *NF- κ B is a transcription factor, which regulates VCAM-1 expression in human endothelial cells. During our study we observed that IL-33 induces VCAM1 protein levels*

starting at 60 minutes. Since transcription precedes protein translation, we only examined the involvement of IL-33 on NF- κ B signaling in HRMVECs for 2 hours.

- 5) In Fig. 2g, h, the processing time of IL33 was 6h, rather than 8h which has the largest increasing trend in Fig. 1c. Moreover, in Fig. 3, the authors chose 8h to process the cells. Please indicate the reasons.

Answer: We are sorry for this confusion. We performed new experiments with 8 h time point to make it consistent throughout the manuscript. The new data is now included in the revised manuscript (**please refer to Fig. 2g and h**).

- 6) CXCL6 is also the homolog of IL-8, and the increased foldchange in Fig. 7b is higher than CXCL1. Therefore, it is suggested to validate CXCL6 by QRT-PCR.

Answer: We agree with Reviewer #2 that CXCL6 is a murine homolog of human IL-8, although no increase in CXCL6 expression was seen in Fig 7b. The reviewer might be referring to Cxcr6 or CXCL2. CXCL2 is a functional murine homolog of IL-8. In Fig. 7b, our RNA sequencing data revealed an enhanced expression of CXCL2. We already performed QRT-PCR experiments to confirm the increased CXCL2 expression by OIR. However, the expression values were not normally distributed between the animals, and we could not detect a significant increase in CXCL2 expression by OIR.

- 7) The study used intravitreal injection at P12 and P14. To inject twice seems to have strong experimental uncertainty which may affect the in vivo results in the OIR model.

Answer: Reviewer #3 has asked a valid question. The half-life of siRNA in the retina depends on the delivery system used (JCI, 2007, 117(12), 3623–3632). In our lab, we have standardized the procedure for delivering siRNA intravitreally, and we have noted that the half-life of siRNA in the posterior portion of the eye is around 24-48 hours. To maintain the impact of siRNA until P17, we administered the siRNA at P12 and P14. For OIR experiments, various groups have employed multiple intravitreal injections of siRNA to observe the effect of a gene downregulation on pathological retinal neovascularization (Blood 2010, 116:1377-1385; Blood 2013, 121:1911-1923; EBioMedicine 2015, 2:1767-1784; Commun Biol. 2022, 5: 479).

- 8) It is also necessary to explore and discuss the relevance of VEGF and VCAM-1 in retinal neovascularization. For example, the authors may check the expressions of VEGFA in some experiments.

Answer: As per Reviewer #3 suggestions, we investigated the effects of IL-33-VCAM1-JunB-IL-8 signaling on VEGFA expression and discovered that IL-33-induced VCAM1-JunB-IL-8 signaling did not affect VEGFA expression in retinal endothelial cells (**please refer to Fig. 1g, Fig. 3e, Fig. 4a, and Fig5f**). The importance of IL-33-induced VCAM1-JunB-IL-8 signaling on VEGF has now been discussed in both the results and discussion

sections (**please refer to page 5, lines 196-198; page 6, lines 201-203; page 11, lines 401-405**).

- 9) In Fig. 8b, c, d, it seems that the retinas were collected at P15 from figure legends, while the results part mentioned they used the results at P17, which is quite chaotic.

Answer: *The retinas were collected at P17 in Fig. 8b, c, and d, but we forgot to mention it in the figure legend. It is now detailed in the revised manuscript's figure legend (**please refer to page 33, line 1230**).*

REVIEWERS' COMMENTS:

Reviewer #2 (Remarks to the Author):

The paper have been revised according to the suggestions. I endorsed the publication of the paper.

Reviewer #3 (Remarks to the Author):

The manuscript has been improved according to the reviewer's suggestion.